

# Removal of Dyes from Simulated Wastewater using Low Cost Activated Carbon Derived from Date Pits

Salam A. Mohammed[1]*, Fazal Mabood[2], Warda Abdlatef[1], Iman Wadi[1], Emad Yousif[3], Ali Abd Ali[3]

[1]Department of Chemical and Petrochemical Engineering, College of Engineering and Architecture, University of Nizwa, 616 Nizwa, Sultanate of Oman.

[2]Department of Chemistry, College of Science, University of Nizwa, 616 Nizwa, Sultanate of Oman.

[3]Department of Chemistry, College of Science, Al-Nahrain University, Baghdad, Iraq

## Abstract

There have been a lot of concerns regarding the pollution in aquatic resources. Since then, there has been a remarkable scientific work in order to remove all sorts of pollutants and offer a reasonably clean environment. In this effort, we show synthesis and characterization of activated carbon (AC) from date pits by various thermal treatments and two different porosities. Furthermore, we demonstrate the removal of four hazardous dyes from simulated waste water via adsorption using three packed bed column as semi batch process. The adsorption experiments demonstrated smooth running flow for the threated water and good removal efficiency for all dyes with some variations. These variations will be adequately displayed and discussed.

Keyword: Low cost activated carbon; dates pits

## Introduction

Dyes have been largely employed in different disciplines especially in textile industry (QiangGao et al., 2017). These dyes are often disposed into water resources from industrial amenities with no further treatment (Parvathi C et al., 2011). Hence, it is necessary to remove these dyes from these water resources in order to alleviate the any catestrophes to aquatic lives as well as human beings (Ramaraju B. et al., 2014, El-Demerdash et al., 2015, Kunwar P. et al., 2005). In this regard, so far, there have been considerable efforts to get rid off these dyes from wastewater via employing various techniques, such as coagulation (Víctor López-Grimau et al., 2015, Kabdaşlı I., 2012, Gilpavas E. et al., 2011), precipitation or flocculation and adsorption (Bolong N et al., 2009). The latter has been found one of the most effective techiques to remove dyes and coloarants from wastewater particularly if there was a chemical or physical interaction between adsorption partners, i.e. adsorbent and adsorbed (Bolong N et al., 2009, Gupta VK et al., 2009). During the past few decades, there have been numerous studies revealed a very efficient adsorption removal of some colorant materials from waters using commercially available and eco-friendly adsorbents, for instance, rice husk, orange peel, and lemon peel (Gupta VK et al., 2009, Ramaraju B et al., 2014). One of these commerciallly available and low-cost adsobents is activated

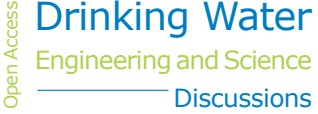

carbon (El-Demerdash et al., 2015).  A large number of some agricultural wastes of cellulosic backbone have displayed their high capacity to eradicate dyes. Dyes like methylene blue, methyl orange has been effectively removed using activated carbon developed from coconut shell fibres (Kunwar P. et al., 2005) and they showed a very high adsorption capacity even at high adsorbate concentrations.

It has been found in the literature that the activated carbon particle size plays a significant role on the adsorption of dyes and consequently on removal of these dyes from water (Hameed and Auta M, 2009).

In a recent study, it has been shown that carbonization temperatures can enhance the surface area of activated carbon materials. Additionally, it has been demonstrated that the microwave heating can produce even higher surface area activated carbon in compared to conventional heating methods (Tamer M. et al., 2013, Natalia and Ekaterina 2015).

In this article, we show the production of activated carbon (AC) from one of commercially available agricultural waste, so-called date pits. These date pits conversion into AC was assisted via two heating instruments, namely, microwave, and furnace. Moreover, we compare the adsorption capacity of AC produced by the above heating methods in removing four dyes, methylene blue (MB), methyl orange (MO), congo red (CR), and eosin yellowish (EY). This study provides some useful information regarding the usage of low-cost and natural waste products in removing pollutants from aquatic sources.

## Experimental
### Materials and Method
All chemicals were sourced from Sigma-Aldrich and used as received.

### Simulated Dye Solutions
A stock solution of 1000 mg/L of each dye was prepared using ultra pure water as a solvent. Thereafter, a series of solutions of different proportions (0 %, 1 %, 3 %, 5 %, 7 %, 10 %, 15 %, 20 %, 25 %, 30 %, 35 %, 40 %, 45 %, 50 %, 55 %, 60 %, 65 %, 70 %, and 75 %) were prepared by sequential dilution. These solutions were actually representing a simulated wastewater with dyes contaminants.

### Preparation of Activated Carbon (AC)
Date pits were obtained from a dates packaging factory at AlSharqiyah, Sultanate of Oman. The pits were thoroughly washed with tap water followed by ultrapure water and then let to dry at room temperature overnight. Thereafter, they were mixed with concentrated sulphuric acid and left for 24 hours, followed by washing few times with copious amounts of ultrapure water. After drying at 70 $^\circ$C, the product was splitted into two portions. The first was burnt in the furnace at 400 C$^\circ$ for two hours under nitrogen flow. The second one was burnt in the microwave oven at medium high energy for 20 minutes under nitrogen flow. After cooling, the two portions were grinded and sieved into a uniform size. The particle size was between (250-425μm) and the second one was between (425-600μm) then kept in clean and sealed jars until use.

### Adsorption Experiments



A series of three fixed beds adsorption column was employed to study the adsorption capacity of
AC to remove the aforementioned dyes from simulated wastewater. All the adsorption tests were
conducted in continuous downward flow mode to keep the head pressure. The experiment was carried
out at room temperature without any pH adjustment. The AC was dried in advance at 70 C° for 2h.
Thereafter, the simulated wastewater of each dye was passed through the padding in a burette and the
time needed to collect every 10 mL of sample from the burette was recorded.
**Characterization of Dyes Solutions**
The dye solutions prior and after the adsorption experiments were characterized using Near Infra
Red spectroscopy (NIR). The Frontier NIR spectrophotometer (BSEN60825-1:2007) by Perkin Elmer
was employed to measure the absorption of all the dye solutions in the wavenumber range from 10000
$cm^{-1}$ to 4000 $cm^{-1}$. Each measurement was done in a triplicate to imporve the reliability.
**Results and Discussion**
The AC was obtained from different thermal treatments, namely microwave, and furnace. In addition,
there were some physical and chemical modifications to improve the porosity and the morphology of
AC as depicted in Figure 1.

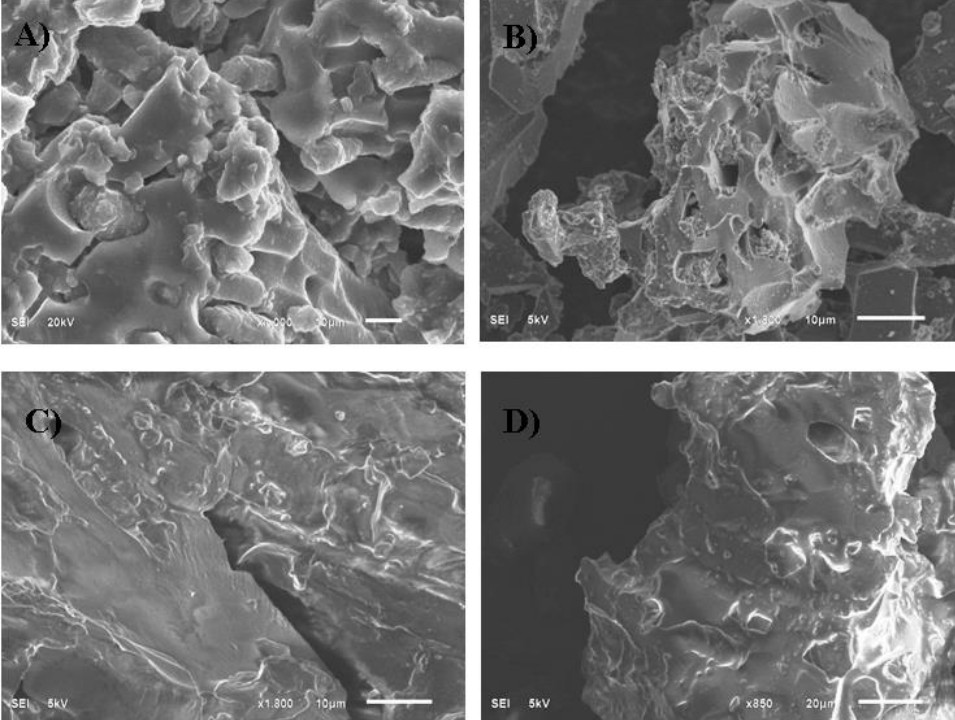

Figure 1. SEM micrographs of A) AC prepared from microwave with physical-chemical modification,
B) AC prepared from furnace with physical-chemical modification, C) AC prepared from furnace with

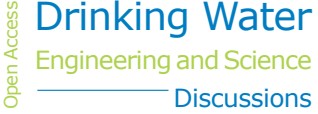



chemical-physical modification, and D) AC prepared from microwave with chemical-physical modification.

Based on the SEM images, we observed that all AC materials prepared showed a very interesting morphological characteristics. The porosity, number of pores, and size of pore were varied based on the thermal treatment. However, in all of the prepared ACs we believe that their porosity is definitely good enough to make them reasonably efficient adsorbents.

Adsorption experiment were performed to test the adsorption capacity of the synthesized AC. The near NIR spectra (Supplementary information, S1) for all of the dyes were collected before and after the adsorption and the concentration was determined from the calibration curve obtained for each dye solution. The goodness of calibration curves (Supplementary information, S2) were judged by the value of $R^2$ which were within the range of (0.80-0.98). There was some light scattering in the spectra collected after the adsorption which might be attributed to the existence of some AC particles in the dye solution.

To imporove the quality an reliability of data obtained and consquently on the accuracy of removal percentage of the dye, we used the first derivative spectra, principal score plots (PCA) and partial least square discriminant analysis (PLS-DA). All of these statistical plots and tables are in the supplementary information, S3.

The efficiency of dyes removal was calculated from the following formula.

$$\% \text{ of Removal} = [\text{dye}]_{initial} - [\text{dye}]_{predicted} \times 100 \tag{1}$$

The predicted concentration of dyes was plotted against time needed to reach the steady state or the saturation level for AC of (250-425 µm), and (425-600 µm) pore size as depicted in Figure 2, and 3.

The time needed to reach the saturation was varied from dye into another and it is probably due to the physical/chemical interaction between the dye and the AC.

The AC synthesized using furnace physical-chemical modification showed the highest removal for MO at pore size (250-425 µm). However, the AC synthesized using furnace chemical-physical modification at (425-600 µm) demonstrated the highest dye removal for CR with slight increase of the MO removal using AC synthesized using furnace physical chemical modification as depicted in Figures 2, and 3.

Overall, the highest dye removal was found in smaller pore size AC, which is consistent with previous studies (Kunwar P. et al., 2005). In fact, we ascribe that to the large surface area of smaller pore size AC which consequently means that dyes are more retained on the AC particles.



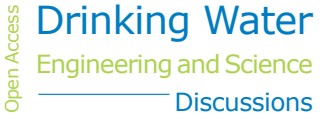

Table 1. summarizes the values of dye removal calculated from equation 1.

| Dye | AC used [a] | Removal (%) | AC used [a] | Removal (%) |
|-----|-------------|-------------|-------------|-------------|
| MO | FCP [c] | 91.91 | FCP | 91.57 |
| CR | = | 90.23 | = | 97.90 |
| MB | = | 64.29 | = | 90.66 |
| EY | = | 56.11 | = | 42.64 |
| MO | MCP [d] | 85.56 | MCP | 91.17 |
| CR | = | 66.75 | = | 72.20 |
| MB | = | 91.01 | = | 83.64 |
| EY | = | 50.22 | = | 45.89 |
| MO | FPC [e] | 94.31 | FPC | 94.31 |
| CR | = | 79.67 | = | 58.88 |
| MB | = | 60.51 | = | 23.14 |
| EY | = | 60.65 | = | 63.61 |
| MO | MPC [f] | 84.4 | MPC | 89.75 |
| CR | = | 76.78 | = | 81.77 |
| MB | = | 40.16 | = | 89.89 |
| EY | = | 59.18 | = | 53.79 |

[a] and [b] are the AC of (250-425), and (425-600 μm) pore size respectively. [c] is the AC from furnace chemical-physical
modification, [d] is AC from microwave chemical-physical modification, [e] AC from physical-chemical modification, and [f] is AC
from microwave physical-chemical modification.

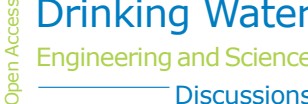



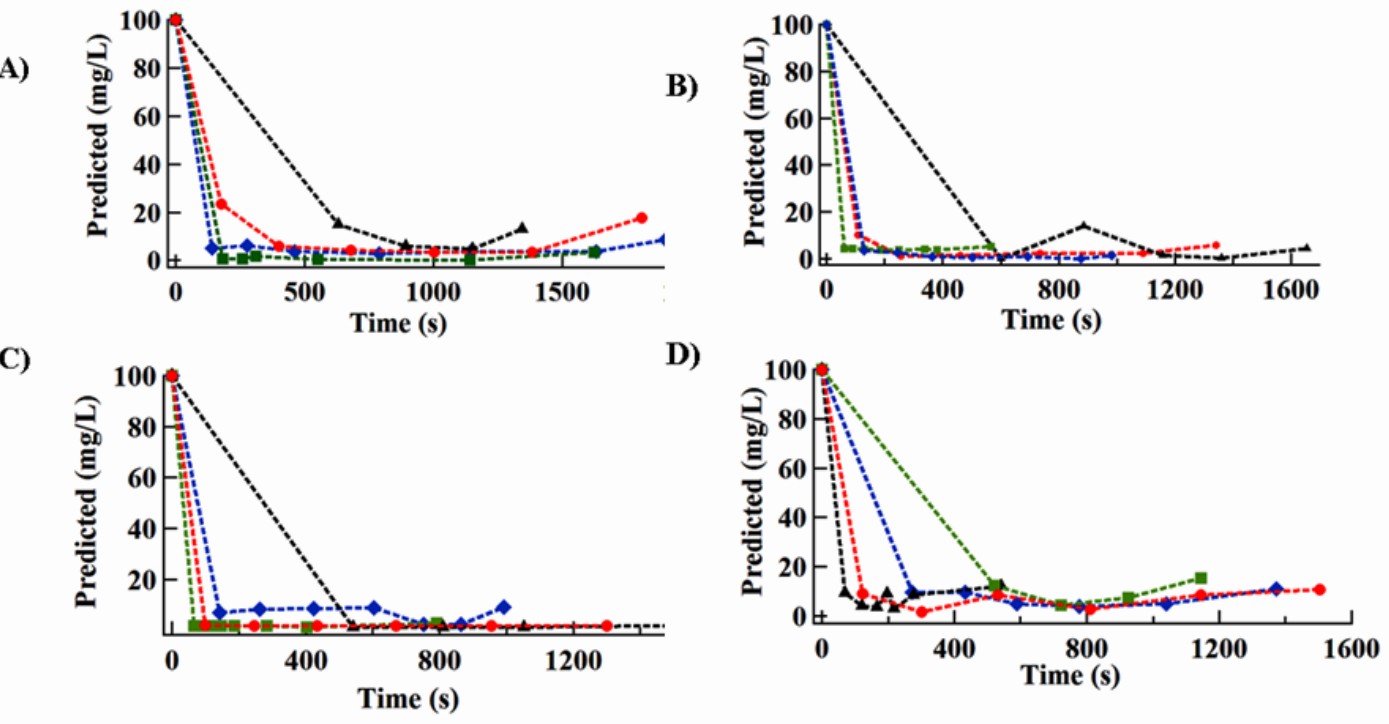

Figure 2. The concentration of the dyes A) methylene blue, B) congo red, C) methyl orange, and D) eyosin yellow removed from the simulated waste water using activated carbon of (250-425 μm) by microwave physical-chemical modification (red circles), furnace physical-chemical modification (black triangles), microwave chemical-physical modification (green squares), and furnace chemical physical modification (blue diamonds).





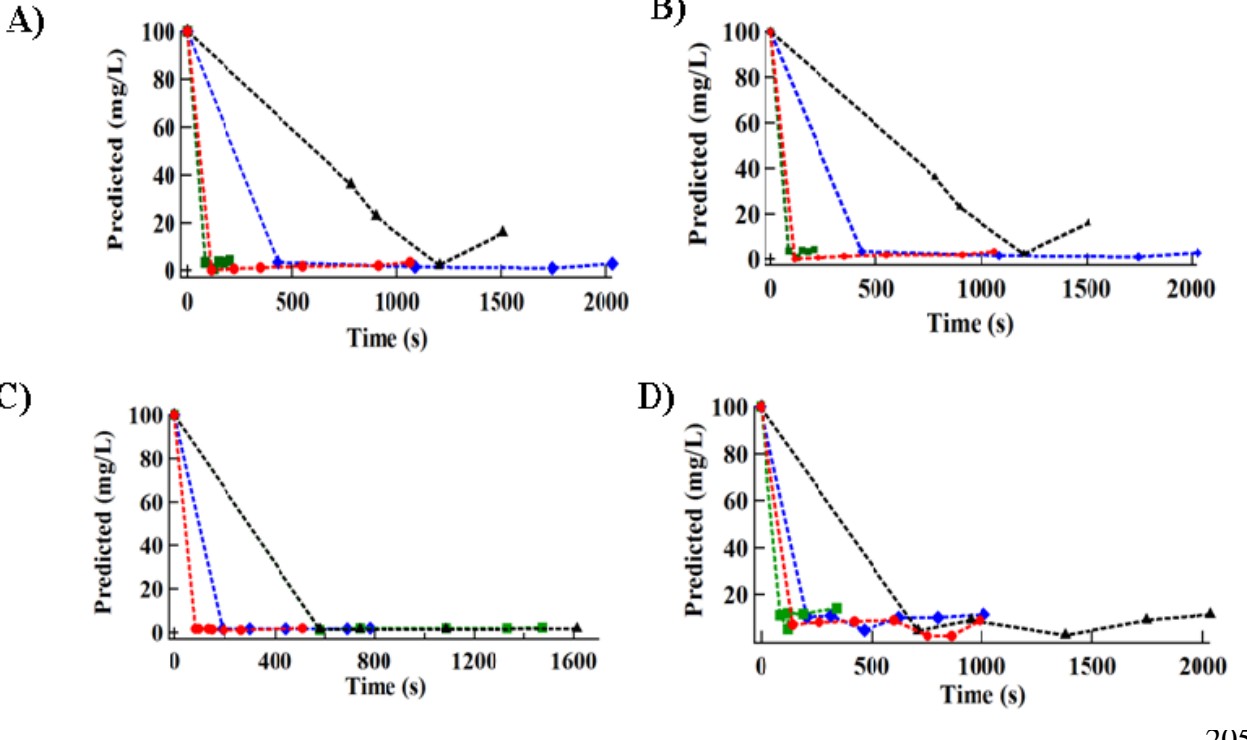

205

Figure 3. The concentration of the dyes A) methylene blue, B) congo red, C) methyl orange, and D) eyosin yellow removed from the simulated waste water using activated carbon of (425-600 μm) by microwave physical-chemical modification (red circles), furnace physical-chemical modification (black triangles), microwave chemical-physical modification (green squares), and furnace chemical physical modification (blue diamonds).

The only dye which showed the least removal was EY as in both AC pore size the maximum removal was ~63%. In this regard, we think that EY was not interacting physically or chemically with the AC, that is why it was less retained in the adsorption column.

**Conclusion**

In summary, the AC synthesized from the date pits exhibited its powerful capacity towards removing various hazardous dye pollutants. These dyes were reasonably adsorbed on the surface of AC of two different porosity. The large surface area of the AC allowed higher amounts of dyes to be adsorbed and get removed. We recommend following the same approach that we presented to remove similar types of dyes and pigments which exist in some wastewater especially from fabrics, and textile industrial plants.

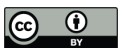

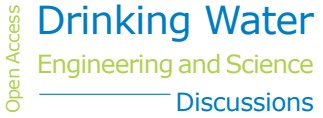

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
