# Peer review of "Removal of Dyes from Simulated Wastewater using Low Cost Activated Carbon Derived from 2 Date Pits 3 Salam A. Mohammed1\*, Fazal Mabood2, Warda Abdlatef1, Iman Wadi1, Emad Yousif3, Ali Abd 4 Ali3 5 6 7 1Departmen"

_Drinking Water Engineering and Science, 2018_

## Referee Comment (RC1) · Anonymous Referee #1 · 5 Mar 2018

DWES reviewer comments

Removal of Dyes from Simulated Wastewater using Low Cost Activated Carbon Derived from Date Pits.

The authors investigated the adsorption of four different dyes on activated carbon prepared from date pits. Waste materials are indeed a cheap precursor for activated carbon as compared to coal/wood/peat based on material costs. The main costs for operating an activated carbon treatment process are regeneration costs. If it would turn out that activated carbons based on waste products can't be effectively regenerated (e.g. high carbon losses, high capacity losses), it may actually turn out to be a

more expensive treatment alternative. So "low cost" can be debated.

In the results and discussion section, it would be helpful to compare the results with other literature. Methylene Blue, for example, is widely used to test activated carbon performance, including activated carbons based on waste materials.

The manuscript is generic at the moment. It could be improved by elaborate more on the experimental choices that were made. For example, both furnace heating and microwave heating were used. Based on literature, the authors could mention what differences are to be expected on the final properties of the activated carbon, and check for these differences, e.g. with a pore size distribution / BET surface analysis. SEM microscopy alone is not sufficient. Then there were 4 different dyes used by the authors, which probably have different size, charge, hydrophobicity. It would be stronger to relate these properties to the differences in adsorption efficacy, rather than just mentioning which one adsorbs well and which one adsorbs poorly.

Specific comments on the manuscript are mentioned below.

Line 18-20: The authors mention aqueous pollution in general as a problem that can be solved with activated carbon, but are specific in their title (i.e. Dyes). It would be better to focus on dyes specifically in the abstract as well.

Line 21: Date pits are introduced as precursor for activated carbon. Could the authors motivate this selection over conventional precursors?

Line 25-26: "These variations...discussed". According to the reviewer, this statement can be omitted. Every researcher tries to display and explain his/her data as good as possible, that doesn't have to be mentioned explicitly.

Line 40-41: Are the authors referring to the direct use of rise husk, orange/lemon peel as adsorbents, or their use as (cheap) precursor for activated carbon?

Line 47-49: Particle size is expected to be important for removal kinetics. Equilibrium adsorption capacity should be closely related to adsorption surface.

Line 56-57: The authors should elaborate on the expected effect of the different heating methods on their activated carbon properties, in order to justify both methods.

Line 58-60: The authors mention four different dyes that are used in their experiments. It would be helpful to explain why these four dyes were selected, again to justify their inclusion.

Line 73: Demiwater with added Dye may not be a good representative for simulated wastewater, as effects of e.g. competition with / pore blockage by NOM, pH value and buffering, and particle fouling in a packed carbon bed are not included. Also, simulated wastewater should contain dye concentrations similar to actual wastewater of e.g. a textile industry case study. It would be more prudent to not refer to your solution as simulated wastewater.

Line 78-80: Acid is also used to activate precursors, and creates an adsorption surface with different functional groups that which are produced when e.g. steam is used for activation. Was the acid used for activation in this work as well? And is this choice expected to be beneficial for dye adsorption? i.e. are functional groups created with acid activation that interact with the dyes?

Line 87: The section on adsorption experiments should describe the amount (mass) of activated carbon that was used, and the filtration velocity or bed volumes treated during continuous down flow treatment.

Line 92: What was the actual pH or the solution? Are the pH probes used sensitive enough in demiwater? Typical pH probes need some conductivity to measure the pH, and demiwater has low conductivity.

Line 92: It is mentioned in the section "preparation of activated carbon" that it was dried at 70 degrees Celcius. Is this a second drying step, or are the authors referring to the same drying step? It might be more clear to include the one, or both drying steps in the "preparation of activated carbon" section.

Line 97-101: When measuring dye concentrations with a spectrophotometer, usually each dye has a characteristic peak at a unique wavelength. The adsorption at that wavelength is correlated to the dye concentration via a calibration curve. Did the authors use the complete wavelength range to measure dye concentrations, or did they use specific wavelengths? If so, please report these for each dye.

Line 104-107: The authors are not clear in describing their method. The activated carbon preparation is described as an acid wash, followed by either heating by furnace or microwave. After this, the carbon is sieved and two size fractions are obtained. Here, in the results and discussion section, an additional modification is mentioned, but it is unclear what exactly this modification is. This should be more explicit, and it should be mentioned in the materials and methods section.

Line 116-120: Do the authors have supplemental analyses to support this, besides only a SEM image? BET surface and/or pore volume distribution analyses would be helpful. A SEM image alone is insufficient to state beforehand that the porosity is enough to have efficient AC.

Line 136: % removal is with respect to the initial concentration, and should be calculated as ( ( [dye] initial (influent) – [dye] final (effluent) ) / [dye] initial (influent) ) * 100. It is unclear why [dye] predicted is used. It is not clear to the reviewer that the authors are using predictive models. If so, equation (1) would probably be Error (%) or Deviation (%), rather than Removal (%), with ( [dye] removal, measured – [dye] removal, predicted) * 100

Line 148: a smaller AC particle size should enhance adsorption kinetics, but eventually result in the same equilibrium adsorption as larger AC particles (all else being equal). Apparently, in the experimental setup of the authors, kinetics are important. In drinking water treatment, typically the equilibrium adsorption capacity is leading, and kinetics are less important.

Line 149-150: For regular activated carbon, the internal adsorption surface is much

larger than the external surface. So, while grinding to smaller diameters does create a larger external surface, this is not by definition significant.

It is unclear how table 1 is connected to Figure 2 & 3. If it is conveying the similar information, the authors should choose using only the figure or only the table. However, in the figures, all dyes seem to be almost completely removed eventually, while table 1 shows much lower removal percentages for EY.

Line 32: spelling; Catastrophes Line 33: spelling; get rid of Line 36: spelling; Colorants (American spelling) or Colourants (British spelling) Line 123: NIR = Near InfraRed. "Near" is already included in the abbreviation. Line 130: "an" should be "and". Other spelling error: consequently. Line 138, line 144, line 148: "pore size" should be "particle size"

Figure 1: contrast of A) / B) / C) and D) within figures is a bit poor.

Table 1 is not referred to, or discussed in the main manuscript. Table 1: the 4th column should have superscript "b" at "AC used". Table 1: FCP/MCP/FCP/MCP abbreviations should be written in full somewhere in the manuscript.

Figure 2 & 3: Y axis title "predicted" is confusing. Are these not experimental results? Figure 2 & 3: The time scale should be the same for all figures and expression in minutes would be more convenient.

---

## Author Comment (AC1) · 8 Mar 2018

Dear Dr We do appreciate your kind constructive comments and we are working on finalizing our response and will provide our reply for each comment been mentioned in two days time

Thanks King regards

---

## Referee Comment (RC2) · Anonymous Referee #2 · 12 Mar 2018

The authors study the transformation of date pits into activated carbon by two different methods and investigate their application as adsorbent to remove four dyes from wastewater. This approach produces a high profitable recycled material and is one of the most merging, ecological and low cost R&D techniques explored by scientific community.

In the introduction section, the authors should to highlight their motivation; on the basis of literature; to use: i) both methods: thermal treatment in a microwave device or in a furnace (give examples of specific surface area values from literature...etc), ii) in chemical activation before thermal treatment, the choice of sulphuric acid as an

impregnant agent (to precise its role, and to expect its benefits compared to others chemical agents such as H3PO4, ZnCl2, KOH,...etc).

The experimental section ought to be more detailed and clarified by presenting a scheme where authors should indicate every step of preparation of the eight AC samples and precise differences between chemical-physical and physical-chemical treatments.

The results as presented in the paper aren't adequately discussed. Authors should make an unambiguous comparison between all AC samples with clear interpretations by i) exploring SEM images, ii) adding N2 adsorption isotherm of each AC (if it is possible) in order to find out the total pore volume, the specific surface area using BET method, and pore size distribution (micropores, mesopores), or to expect the values of these parameters on the basis of literature results (if it is not possible), iii) making adsorption experiments using different initial concentrations of dyes to find out the adsorption isotherm of each dye and to fit it by usual models (BET, Freundlich,...etc). These would be helpful to conclude about interactions between each dye and adsorbent.

Comments on the manuscript:

- The authors (Ath) should mention in the abstract and introduction that the precursor (date pits) was impregnated firstly by sulphuric acid before pyrolysis in furnace or microwave oven.

- In the experimental procedure, (Ath) should precise the concentration of sulphuric acid, the quantity of date pits impregnated by acid, the ratio of impregnation (weight of acid/weight of pre-treated date pits) and comment the choice of the temperature 400°C for pyrolysis in furnace, precise the power of the microwave oven.

Line 66-67:"...different proportions, (Ath) should avoid repetition of the unit % - Different concentrations of dyes were prepared by authors as explained in experimental

paragraph. Authors have to precise that those solutions served to establish the calibration curve for NIR measurements and only the concentration 100mg/L of each dye was used in adsorption experiments.

Lines 75 & 76: (Ath) should replace "burnt" by "pyrolyzed".

Line 77: the statement is not clear: After cooling, the two portions were grinded and sieved into a uniform size. The particle size was between (250-425$\mu$m) and the second one was between (425-600$\mu$m) Correction: After cooling, each portion was grinded and sieved into two size distributions. The particle size of the first distribution was between (250-425$\mu$m) and for the second one was between (425-600$\mu$m).

- (Ath) should add a statement where they attribute the names for each sample: for example FAC1 and FAC2 for activated carbon (AC) particles prepared by pyrolysis in furnace and having size range 250-425$\mu$m, and 425-600$\mu$m respectively.MAC1 and MAC2 : for AC pyrolyzed in microwave oven and having size distribution as for FAC1 and FAC2 respectively. . .

Line 82: A series of three fixed beds adsorption column: four dyes have been tested: MB, MO, CR, and EY, and until this line, it is clear that (Ath) prepared 4 samples of AC, however, eight samples are tested in the section of results. -In adsorption experiments: the weight of each AC sample used as fixed bed for adsorption should be specified.

Lines 92-93: (Ath) should explain and precise if they use a peak at a unique wavenumber for adsorption kinetic monitoring or all obtained absorbance peaks in the range 4000 -10000 cm-1

Line 97-98: the statement is not clear; explain the physical and chemical modifications that have been made to improve the porosity and the morphology of AC.

Line 103: caption of Figure1: did (Ath) mean SEM micrographs of A) MAC1, B) FAC1, C) FAC2? If yes, make the necessary corrections. If not, (Ath) should add precisions in the experimental paragraph (preparation of activated carbon) about difference between

physical-chemical modifications and chemical-physical modifications.

Line 109: (Ath) should describe the size distribution of pores obtained according to SEM images for comparison (only macropores are observed by this technique) to make evidence that number of pores, and their size were varied based on the thermal treatment, it is recommended to authors to measure N2 adsorption isotherms of their samples for making a precise assessment (pore volume and pore size distributions: micropores, mesopores).

Line 113-117: Authors should add a Figure presenting some NIR absorbance spectra (for illustration) (for example: at the beginning of adsorption, after 100s, and at the end of adsorption for MAC1 and FAC1).

Lines 125, 126: in equation (1): correct the formula of

% removal = {([dye]initial − [dye]final)/[dye]initial}×100

(Ath) should replace "Predicted concentration" by "concentrations of dyes deduced from NIR absorbance measurements was plotted against time until reaching the adsorption equilibrium for FAC1, MAC1,…as depicted in figures 2 and 3". Omit "pore size" in the statement.

Line 128: The time needed …was varied from…: omit "was varied" and substitute it by: "was different"

Line 131: the interval (250-425$\mu$m) is the range of AC particle size not the range of pore size since Authors didn't present any results about pore size distributions of AC samples. There is an ambiguity in results, as mentioned in experimental paragraph by authors, fours AC samples were prepared by chemical activation followed by heat treatment in furnace or in microwave oven, then the two samples were divided into two parts according to the range of their size after grinding and sieving. However, in figure 2 and 3 and in table 1, authors report results about eight samples.

Line 132: ..at pore size (250-425$\mu$m) : omit "pore size"

Line 133 -137: Other usual key parameter to use for comparison is the adsorption capacity of each AC sample (grams of adsorbed dyes/grams of AC). The initial rate of adsorption (to calculate from the first two measured concentrations of dye during adsorption test) could be also useful.

Line 135: "Overall, the highest dye removal was found in smaller pore size AC": (Ath) should omit this statement, because it is not true. There is no clear correlation between size of AC particles (not pore size) which refers to the external accessible surface and the % removal of dyes.  % removal or adsorption capacity (which is related to the internal surface accessibility of AC) depend on the pore size distribution, total pore volume (micropores and mesopores), specific surface area SBET, physical (physic-sorption) and/or chemical (chemi-sorption) interactions between adsorbent and dye, size and shape of dye molecules...etc and all these parameters aren't measured for instance in this paper.

Line 212 to 214: For EY removal, the explanation of the result is generic. (Ath) should give more specific interpretations (on the basis of functional groups of MB molecules, EY, and those expected to have at AC surface samples according to the previous works in the literature).

Table 1 must be cited in the text.

Figures 2 and table1: there is a discrepancy in % removal indicated in table 1 and the results of figure 2.a: in figure 2.a, the % removal of MCP and FCP are the highest and are close, but in Table 1 (column a) The highest value calculated from figure2.a corresponds to FPCe, moreover, %removal of FCPc and MCPd are different.  (Ath) should check again their results for more consistency.

Figures 2 and 3: It is more convenient to plot (Concentration of adsorbed dye versus time) where: Concentration of adsorbed dye = Initial concentration – measured concentration in the solution) (mg/L)

Line 219: (Ath) should omit "two different porosities" and substitute it by "two different size particles", -The conclusion is generic; (Ath) should give more details of the obtained results.
* * *

---

## Author Comment (AC2) · 13 Mar 2018

Our sincere thanks for the efforts by valued reviewer for the constructive comments Responses to the reviewer comments We will take each point been mentioned and here we are providing our response individually: Generally all the comments related to the abstract will be implemented in the final version of the paper (Lines 18-20, 21, 18-28). Line 40-41 quite number of researchers has studied those materials as adsorbent to replace activated carbon (AC).

Line 47-49: true enough the point been mentioned and we totally agree with, therefor we chosen two ranges of the particles sizes (250-425, 425-600 $\mu$m) in line 78.

Line 58-60: we have selected four different types of organic dyes to study the performance of the generated AC on the dyes removal.

Line 73: In fact, we are targeting to use the actual textile wastewater in our second level of the research after we have proven the generated AC on dyes removal from the simulated wastewater. We strongly support this comment.

Line 78-80: Both activations were implemented. Chemical activating was conducted (line 73) and then thermal activation was been done.

Line 87: the total mass been used in the three packed beds will be mentioned in our final version of the article but in general 5 ml as volume been filled by AC in each stage (so total volume was 15 ml as whole process).

Line 92: pH was measured using calibrated. Every time before the measuring the calibration was performed using three different standard solutions (pH: 4, 6, and 8).

Line 92: here we believed that the respective reviewer referring to line 82 instead of, this heating as performed on the final AC produced before conducting the water treatment insure that we have exact starting condition for the experiment for each run.

Line 97-101: surly we have generated the calibration curve and table 1 results were based on that particular curve. It will be added in the final draft of the paper.

Line 104-107: This part of the result will be discussed in more details in the final draft. Line 136: the removal % results were conducted automatically using NIR software (Unscrambler Portable)

Line 148: this point is true at certain conditions, but we believe that when the pollutant is kind of martials having high surface tension then the behavior will be different. We have found in our current second level of this research that the best efficiency at particle size between certain range in compare with other probability which we will show it the coming new article.

Line 149: we totally agree. Graphs 2 and 3 they are note same and are presenting two different particle sizes with the dyes removal results. The required changes on the X and Y axis format will be done and replace them in the final version of the article. With regards to table 1 and graphs 2 and 3, we believe that both are important as we would like to show readers how significant the decrease in the percentage of the removal was using the numbers in the table. Some people can follow that up easily using the table. However, graphical representation is also very useful and gives a clear idea about the adsorption, and the amount of reduction in the dye concentration. If we are required to choose between figure and table, we would certainly remove the table and keep the figure. At the end, as main target of this research was to compare the AC generated using ordinary commercial microwave efficiency with its generated using furnace as power saving and time need for the activation. Salam A. Mohammed (Ph.D)

–––––––––––––––––––––––––––––––

---

## Author Comment (AC3) · 18 Mar 2018

Dear valued Dr

We are thankful for your compromising in the introduction of your valuable comments We totally agree with all the points been raised and we will perform all the responses needed accordingly the final version of the article. Regarding the BET test, we have sent our samples to one of labs which has the apparatus to conduct the test (which we expect that the result will be out after one week) and we believe that this test will answer most of the comment and questions related specially regarding the surface area, porosity and particles distribution. For the microwave power used, was 800 Watt

We will include all these details in the final version

Thanks again
* * *

---

## Author Comment (AC4) · 16 Apr 2018

Responses to the reviewer comments In general, our intention of this research is to treat waste water using some of agriculture waste and re generate the consumed AC is not in our objective because as it is been indicated by valued reviewer it is expensive and it will not be economic. Treating waste water with cheap AC from some waste in order to have water with less pollutant is sustainable green technique We will take each point been mentioned and here we are providing our response individually: Generally all the comments related to the abstract will be implemented in the final version of the paper (Lines 18-20, 21, 18-28) and the revised version of the abstract became as the

following: [There have been a lot of concerns regarding the pollution in aquatic resources. Since then, there has been a remarkable scientific work in order to remove all sorts of pollutants (as organic and inorganic components) and offer a reasonably clean environment. In this effort, we show synthesis and characterization of activated carbon (AC) from date pits using various thermal activation (ordinary furnace and commercial microwave) and then chemical activation using concentrated H2SO4. Moreover, another series of samples were generated by chemical activation and followed with thermal activation. Furthermore, we demonstrate the removal of four hazardous dyes as organic pollutants from simulated waste water via adsorption using three packed bed column as semi batch process. The adsorption experiments demonstrated smooth running flow for the threated water and good removal efficiency for all dyes with some variations. The highest performance of AC reached up to 97.9% and the efficiency variations will be adequately displayed and discussed.] Line 40-41 quite number of researchers has studied those materials (such a rice husk, coconut shell fiber etc) as adsorbent to replace activated carbon (AC) and a sentence has been added into the text related to this point. Line 47-49: true enough the point been mentioned and we totally agree with, therefor we chosen two ranges of the particles sizes (250-425, 425-600 $\mu$m) in line 78. Paragraph been added to highlight it Line 58-60: we have selected four different types of organic dyes (as wide range of dyes available) to study the performance of the generated AC on the dyes removal. This sentence was paraphrased in a way to demonstrate the reason behind of dyes selection. Line 73: In fact, we are targeting to use the actual textile wastewater in our second level of the research after we have proven the generated AC on dyes removal from the simulated wastewater. We strongly support this comment. Line 78-80: Both activations were implemented. Chemical activating was conducted (line 73) and then thermal activation was been done. Line 87: A total mass of 6 gm of AC been used in the three packed beds with 0.083 ml.sec-1 as average filtration velocity for the water flow. Line 92: pH was measured using calibrated pH probe was around (8-8.4 based on the dyes type). Every time before the measuring the calibration was performed using three different standard

solutions (pH: 4, 6, and 8). Line 92: here we believed that the respective reviewer referring to line 82 instead of, this heating was performed on the final AC produced for 1.5 hour before conducting the water treatment insure that we have exact starting condition for the experiment for each run. Line 97-101: surly we have generated the calibration curves for each dye and these curves were added in the text as figure number 2 results were based on that particular curve. Line 104-107: We totally agreed with this comment and this part of the methodology was paraphrased to state the procedure with clear explanation. The results are discussed thoroughly in the final draft. Line 136: the removal % results were conducted automatically using NIR software (Unscrambler Portable) and we have corrected the formula as mentioned in the respective reviewer comment. We had a mistake in writing the relation mentioned. In other hand, basically predicted as term is same as final effluent which indicate to the dye remaining concentration Line 148: this point is true at certain conditions, but we believe that when the pollutant is kind of martials having high surface tension then the behavior will be different. We have found in our current second level of this research that the best efficiency at particle size between certain range in compare with other probability which we will show it the coming new article. Line 149: we totally agree and paragraph was added in the text. Graphs 3 and 3 they are note same and are presenting two different particle sizes with the dyes removal results. The required changes on the X and Y axis format are made and the new figures have replaced the old one. To maintain consistence in the text style we have eliminated table one from the article as has been recommended by respective reviewer. At the end, as main target of this research was to compare the AC generated using ordinary commercial microwave efficiency with its generated using furnace as power saving and time need for the activation.

---

## Author Comment (AC5) · 16 Apr 2018

We are thankful for your compromising in the introduction of your valuable comments. We totally agree with all the points been raised. It is our pleasure to provide the following responses according to the respective reviewer comments: Regarding the introduction part, we have already did some editing brought some other researchers work finding about AC field. Selecting H2SO4 as impregnant agent was based on some researchers' recommendation about its better performance as agent. The methodology section was paraphrased to provide complete details for the work procedures for chemical-physical and then physical-chemical treatment to generate AC. We have dis-

cussed our conducted experiment data thoroughly and BET results for each sample of AC are presented in the final draft as well along the changes mentioned above For the abstract, a sentence added to explain the activation procedure (thermal-chemical, chemical-thermal) were added. Same related point was edited in the introduction section in the research objectives. For the experiment, as chemical activation a 40% concentration sulphuric acid (with mass ratio 1:4 of AC/H2SO4 respectively) and left for 24 hours. Panasonic microwave with 800 watt was used for the thermal treatment. Those information were added in the paper text Line 66-67: % as symbol was removed and just kept at the end as been recommend and the required paragraph to refer to the calibration curve was edited Line 75-76: "burnt" as term was replaced by "pyrolyzed" Line 77: the sentence is been paraphrased accordingly Table one was removed to maintain the article certainty and text flow. We have modified the discussion section to discuss the results based on the figures conducted to minimize the confusing. Line 82: in fact we had 8 samples as total have been tested, true enough four samples but we did divide them into two size distribution they became eight samples. After we edited the methodology part and in the discussion section as well, this point became very clear. For the bed setting, we have modified the part to state clearly that we used fixed bed technique. Line 97-98: what we wanted to mention here is the activation concept, we have replaced this word to be activation to eliminate the confusion. Line 103: a paragraph to explain the mentioned point was added in the research methodology. Line: all the information related to the particle BET test for each sample will be presented in the final version of the article where by the surface area, pore size and distribution data will be summarized in table form. Line 125-, 126: the formula was modified; we do admit there was error in typing the relation. Please do accept our sincerely apologize The suggested changes in the results discussion part were performed in line 126-128 Line 132, 219: "pore" as term was replaced by "size" Line 113-117: the calibration curve for each dye is been added to the article Line 135: we removed this paragraph Line 212-214: we have added further discussion regarding it and supported this discussion with other researcher reports Figures presenting the removal % were modified to display

the data in more clear style as time scale and as well the y axis.